# Double-$J/\psi$ system in the spotlight of recent LHCb data

V. Baru[1,2], X.-K. Dong[3,4], F.-K. Guo[3,4], C. Hanhart[5], A. Nefediev[6*] and B.-S. Zou[3,4,7]

**1** Institut für Theoretische Physik II, Ruhr-Universität Bochum, Bochum D-44780, Germany
**2** Institute for Theoretical and Experimental Physics NRC "Kurchatov Institute",
Moscow 117218, Russia
**3** CAS Key Laboratory of Theoretical Physics, Institute of Theoretical Physics,
Chinese Academy of Sciences, Beijing 100190, China
**4** School of Physical Sciences, University of Chinese Academy of Sciences,
Beijing 100049, China
**5** Institute for Advanced Simulation, Institut für Kernphysik and Jülich Center for Hadron
Physics, Forschungszentrum Jülich, Jülich D-52425, Germany
**6** Jozef Stefan Institute, Jamova 39, 1000, Ljubljana, Slovenia
**7** School of Physics, Central South University, Changsha 410083, China

⋆ a.nefediev@gmail.com

## Abstract

**Recently the LHCb Collaboration announced intriguing results on the double-$J/\psi$ production in proton-proton collisions. A coupled-channel interpretation of the measured di-$J/\psi$ spectrum is presented and a possible nature of the proposed near-threshold state $X(6200)$ is discussed.**

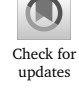

## 1 Introduction

Recently the LHCb Collaboration announced the first measurement of the double-charmonium production in proton-proton collisions [1]. The cross section of the double-$J/\psi$ production was measured (see the left plot in Fig. 1) and a significant ($5\sigma$) deviation from a non-resonant production was found (see the right plot in Fig. 1). In particular, a narrow resonance-like structure at 6.9 GeV and a broad structure just above the double-$J/\psi$ threshold were reported by LHCb. We present a theoretical coupled-channel analysis [2] of the LHCb data and discuss a possible molecular interpretation [3] of the proposed fully charmed tetraquark state residing very near the double-$J/\psi$ threshold (hereinafter referred to as $X(6200)$) entailed from this analysis.

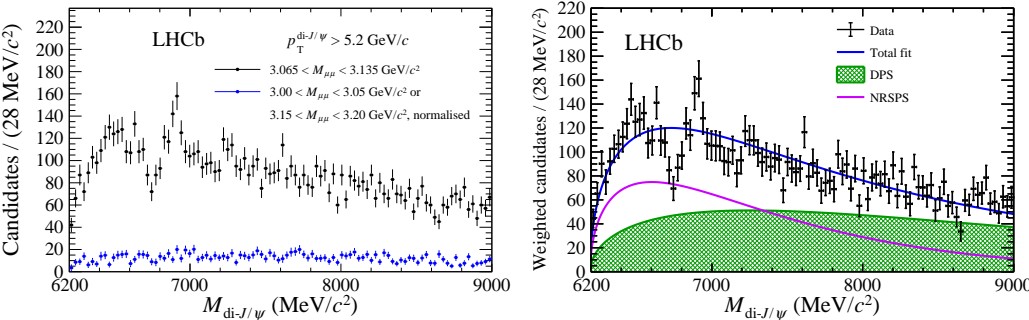

Figure 1: Data on the double-$J/\psi$ production in proton-proton collisions provided by the LHCb Collaboration (left plot) superposed with the non-resonant distributions (right plot) where NRSPS and DPS stand for the NonResonant Single Parton Scattering and Double Parton Scattering, respectively. Adapted from Ref. [1].

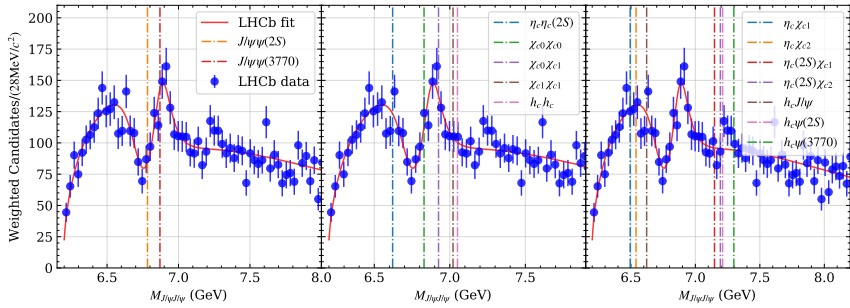

Figure 2: The LHCb data and best fit reported in Ref. [1] superposed with the double-charm thresholds residing in the energy range from 6.2 to 7.2 GeV. Only relevant thresholds are retained in the left plot (see the text for details) while additional (not considered) $S$- and $P$-wave thresholds are given in the middle and right plot, respectively.

## 2 Theoretical data analysis

### 2.1 Coupled-channel approach

A theoretical analysis of the LHCb data reported in Ref. [1] requires an approach based not on naive models and parametrisations such as the Breit-Wigner formula but on a suitable coupled-channel approach since many double-charmonium thresholds reside in the energy range of interest between 6.2 and 7.2 GeV (see Fig. 2). Meanwhile, the quality of the present data does not allow one to reliably fix as many fitting parameters as needed to include all these channels. Therefore, minimal possible models should be employed in the analysis that implies that only the most relevant channels are retained and the minimal necessary order in the Effective Field Theory (EFT) expansion is employed. Thus, as the first step we reduce the number of channels by

- considering only $S$-wave channels with the thresholds lying in the range 6.2-7.2 GeV,

- retaining only transitions from the double-$J/\psi$ channel through light (not heavier than two pions) exchanges; this allows us to disregard all $\chi_{cJ}\chi_{cJ}$ ($J = 0, 1$) channels which can be produced from the $J/\psi J/\psi$ one through $\omega$-exchanges regarded as heavy,

- excluding heavy quark spin symmetry (HQSS)-suppressed transitions between channels, that is, neglecting the transitions like $J/\psi J/\psi \leftrightarrow h_c h_c$ which require a heavy quark spin flip suppressed by the small ratio $\Lambda_{\text{QCD}}/m_c$.

## 2.2 Models

As explained above, we stick to the minimal possible coupled-channel model consistent with the data. In particular, we consider (i) a 2-channel ($J/\psi J/\psi$ & $\psi(2S)J/\psi$) model with the potential

$$V_{2\text{ch}}(E) = \begin{pmatrix} a_1 + b_1 k_1^2 & c \\ c & a_2 + b_2 k_2^2 \end{pmatrix}, \tag{1}$$

containing 5 real parameters, and (ii) a 3-channel ($J/\psi J/\psi$, $\psi(2S)J/\psi$ & $\psi(3770)J/\psi$) model with a 6-parameter potential

$$V_{3\text{ch}}(E) = \begin{pmatrix} a_{11} & a_{12} & a_{13} \\ a_{12} & a_{22} & a_{23} \\ a_{13} & a_{23} & a_{33} \end{pmatrix}. \tag{2}$$

The multichannel amplitude ($T$-matrix) is found as a solution of the Lippmann-Schwinger equation,

$$T(E) = V(E) + T(E) \cdot G(E) \cdot V(E) \implies T(E) = V(E) \cdot [1 - G(E) \cdot V(E)]^{-1}. \tag{3}$$

Here, depending on the version of the model, $V(E)$ is either $V_{2\text{ch}}(E)$ or $V_{3\text{ch}}(E)$, and $G(E)$ is a diagonal matrix of the two-body propagators with the elements [4]

$$G_i(E) = \frac{1}{16\pi^2} \Bigg\{ a(\mu) + \log\frac{m_{i1}^2}{\mu^2} + \frac{m_{i2}^2 - m_{i1}^2 + E^2}{2E^2} \log\frac{m_{i2}^2}{m_{i1}^2} + \frac{k}{E}\Big[ \log\left(2k_i E + E^2 + \Delta_i\right)$$
$$+ \log\left(2k_i E + E^2 - \Delta_i\right) - \log\left(2k_i E - E^2 + \Delta_i\right) - \log\left(2k_i E - E^2 - \Delta_i\right) \Big] \Bigg\}, \ \Delta_i = m_{i1}^2 - m_{i2}^2, \tag{4}$$

where $m_{i1}$ and $m_{i2}$ are the particle masses in the $i$-th channel, $k_i = \lambda^{1/2}(E^2, m_{i1}^2, m_{i2}^2)/(2E)$ is the corresponding three-momentum with $\lambda(x, y, z) = x^2 + y^2 + z^2 - 2xy - 2yz - 2xz$ for the Källén triangle function; $\mu$ denotes the dimensional regularisation scale, and $a(\mu)$ is a subtraction constant. In practical calculations we use $\mu = 1$ GeV and $a(\mu = 1 \text{ GeV}) = -3$, keeping in mind that its variance can be absorbed into the redefinition of the contact interactions in the potential. The $T$-matrix from Eq. (3) respects constraints of unitarity.

Then the production amplitude in the $J/\psi J/\psi$ channel (denoted as channel 1) is built as

$$\mathcal{M}_1 = \alpha e^{-\beta E^2}\Big[ b + G_1(E)T_{11}(E) + G_2(E)T_{21}(E) + r_3 G_3(E)T_{31}(E) \Big],$$

where the slope $\beta = 0.0123$ GeV$^{-2}$ is pre-fixed from the fit to the double-parton scattering (DPS) — see the right plot in Fig. 1, and the parameter $r_3$ is

$$r_3 = \begin{cases} 0 & \text{2-channel model} \\ 1 & \text{3-channel model} \end{cases},$$

so that the production amplitude contains two additional fitting parameters: the overall normalisation $\alpha$ and the background $b$. We, therefore, end up with 2-channel 7-parameter and 3-channel 8-parameter models.

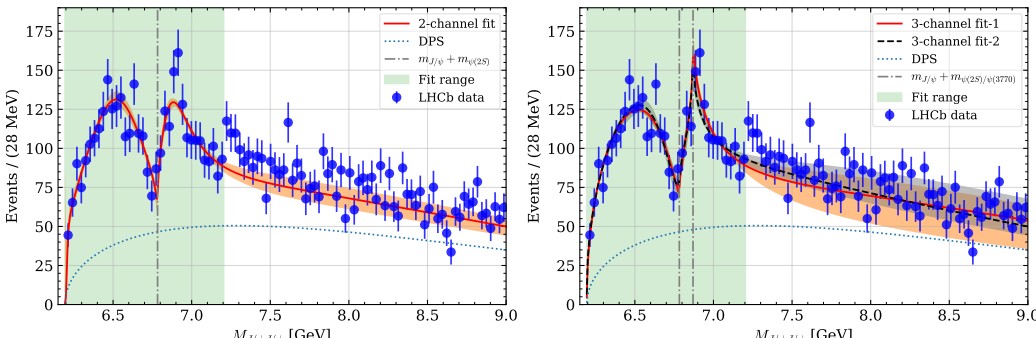

Figure 3: The fitted line shapes for the 2- (left plot) and 3-channel (right plot) model. Adapted from Ref. [2] under arXiv.org non-exclusive license to distribute.

## 2.3  Fit results

The fitted line shapes for the two models described in detail in the previous chapter are shown in Fig. 3 and the corresponding parameters are listed in Tables 1 and 2. It should be noticed that all parameters with bars quoted in the tables need to be multiplied by $\prod_{i=1}^{4} \sqrt{2m_i}$, where $m_i$'s are the involved charmonium masses [5],

$$m_{J/\psi} = 3.0969 \text{ GeV} \quad m_{\psi(2S)} = 3.6861 \text{ GeV}.$$

From the values of $\chi^2/\text{dof}$ quoted in Tables 1 and 2 one can see that all three fits (one fit for the 2-channel model and two fits for the 3-channel model) provide an almost equally good description of the present data, so that the latter do not allow one to discriminate between the two models and different types of description within the same model. Therefore, we highlight a further prediction of our two models — the invariant mass spectrum in the $\psi(2S)J/\psi$ channel — which, if measured experimentally, could allow one to distinguish between them (see Fig. 4).

Meanwhile, comparing the predictions of the two models with each other we interpret only those of them which are robust with respect to the model modification. On the contrary, we refrain from interpreting the results and predictions which appear to be different for different versions of the model. In particular, from Fig. 5, where the position of the poles of the amplitude are shown for all three fits from Tables 1 and 2, one can conclude that the poles above the double-$J/\psi$ threshold, the most prominent of which is known in the literature as the $X(6900)$, are badly determined by the data, so that its parameters (the "mass" and "width" which can be identified with the real part and twice the imaginary part of the pole, respectively) are highly uncertain. On the contrary, all versions of the coupled-channel model employed and all fits found are consistent with the existence of a pole near the double-$J/\psi$ threshold which we refer to as the $X(6200)$ (see Ref. [2] for further details). This finding was confirmed independently in Ref. [6]. Therefore, we regard the existence of this new state as a robust prediction and discussed its possible nature in more detail below. Since the Bose symmetry for the state formed by two identical $J/\psi$ mesons precludes the total spin 1 of this system, the possible quantum numbers of the proposed $X(6200)$ are $J^{PC} = 0^{++}$ or $2^{++}$.

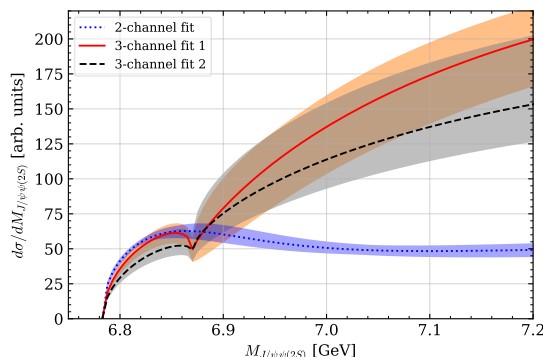

Figure 4: Predictions for the invariant mass spectrum in the $\psi(2S)J/\psi$ final state. Adapted from Ref. [2] under arXiv.org non-exclusive license to distribute.

Table 1: Fitted parameters of the 2-channel model ($[\bar{a}_i]$=GeV$^{-2}$, $[\bar{b}_j]$=GeV$^{-4}$, $[\bar{c}]$=GeV$^{-2}$) and $\chi^2$/dof.

| $\bar{a}_1$ | $\bar{a}_2$ | $\bar{c}$ | $\bar{b}_1$ | $\bar{b}_2$ | $\alpha$ | $b$ | $\chi^2$/dof |
|---|---|---|---|---|---|---|---|
| $0.2^{+0.6}_{-0.5}$ | $-4.2 \pm 0.7$ | $2.94^{+0.36}_{-0.29}$ | $-1.8^{+0.4}_{-0.5}$ | $-7.1 \pm 0.4$ | $70^{+8}_{-7}$ | $3.3 \pm 0.4$ | 0.99 |

Table 2: Fitted parameters of the 3-channel model ($[\bar{a}_{ij}]$=GeV$^{-2}$) and $\chi^2$/dof.

| $\bar{a}_{11}$ | $\bar{a}_{12}$ | $\bar{a}_{13}$ | $\bar{a}_{22}$ | $\bar{a}_{23}$ | $\bar{a}_{33}$ | $\alpha$ | $b$ | $\chi^2$/dof |
|---|---|---|---|---|---|---|---|---|
| $6.0^{+2.2}_{-1.6}$ | $10.3^{+3.4}_{-2.8}$ | $-0.2^{+1.9}_{-1.3}$ | $13^{+5}_{-4}$ | $-2.6^{+2.4}_{-1.3}$ | $-2.3^{+1.5}_{-1.1}$ | $250^{+70}_{-60}$ | $-0.12^{+0.21}_{-0.22}$ | 0.97 |
| $7.8^{+3.4}_{-2.0}$ | $16 \pm 4$ | $0.9^{+2.3}_{-2.5}$ | $26^{+12}_{-6}$ | $-3^{+4}_{-5}$ | $-2.5^{+2.1}_{-1.0}$ | $144^{+67}_{-27}$ | $-0.7^{+0.5}_{-0.4}$ | 1.05 |

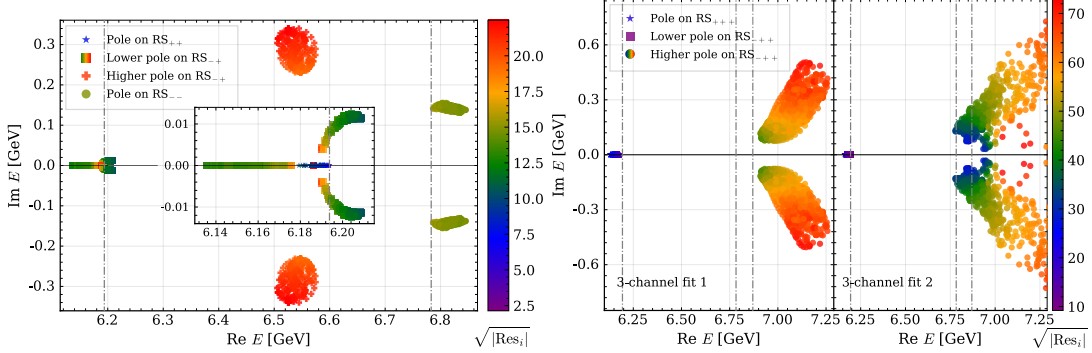

Figure 5: The poles of the amplitude for the 2- (left plot) and 3-channel (right plot) model. Adapted from Ref. [2] under arXiv.org non-exclusive license to distribute.

# 3 $X(6200)$ as a double-$J/\psi$ molecule

## 3.1 Compositeness of the $X(6200)$

It is obvious from the production mechanism that the proposed $X(6200)$, if it exists, must be a fully charmed tetraquark $\bar{c}\bar{c}cc$ state, however the clustering of the quarks may be different: this

Table 3: The effective range parameters in the $J/\psi J/\psi$ channel and the composite-ness $\bar{X}_A$ of the proposed $X(6200)$.

|  | 2-ch. fit | 3-ch. fit 1 | 3-ch. fit 2 |
|---|---|---|---|
| $a_0$(fm) | $\leq -0.49$ or $\geq 0.48$ | $-0.61^{+0.29}_{-0.32}$ | $\leq -0.60$ or $\geq 0.99$ |
| $r_0$(fm) | $-2.18^{+0.66}_{-0.81}$ | $-0.06^{+0.03}_{-0.04}$ | $-0.09^{+0.08}_{-0.05}$ |
| $\bar{X}_A$ | $0.39^{+0.58}_{-0.12}$ | $0.91^{+0.04}_{-0.07}$ | $0.95^{+0.04}_{-0.06}$ |

can be either a compact tetraquark formed by the confining forces of QCD or a weakly bound molecular state formed by soft gluon exchanges between two $J/\psi$ mesons. The developed coupled-channel approach allows us to evaluate the compositeness of this state which defines the probability to observe it in the form of a double-$J/\psi$ system. To this end we define the nonrelativistic scattering amplitude in the $J/\psi$-$J/\psi$ channel,

$$T(k) = -8\pi E \left[ \frac{1}{a_0} + \frac{1}{2} r_0 k^2 - i\,k + \mathcal{O}(k^4) \right]^{-1}, \quad E = 2m_{J/\psi} + \frac{k^2}{m_{J/\psi}}, \tag{5}$$

and extract the values of the scattering length $a_0$ and effective range $r_0$ (see Table 3). By convention, the sign of the scattering length is negative (positive) for the bound (virtual) state.

According to the findings of Ref. [7], the compositeness of the proposed $X(6200)$ is evaluated as

$$\bar{X}_A = (1 + 2|r_0/a_0|)^{-1/2}, \tag{6}$$

and the corresponding numerical values obtained for different fits are quoted in Table 3, from which one can see that the LHCb data on the double-$J/\psi$ production are consistent with $\bar{X}_A \simeq 1$ that hints towards a molecular nature of the $X(6200)$.

## 3.2 Binding forces

It was demonstrated in the previous chapter that, according to the theoretical coupled-channel analysis of the data currently available, there exists a pole near the double-$J/\psi$ threshold which is very likely to be a $J/\psi$-$J/\psi$ molecule. Therefore, a natural question is what interactions between two $J/\psi$'s could produce such a near-threshold pole. It has been known since long ago [8,9] that the interaction between heavy quarkonia mediated by soft gluon exchanges hadronise as light-meson ($\pi\pi, K\bar{K}$) exchanges and can be described in terms of the multipole expansion which is valid for $r_{\bar{Q}Q} \ll \Lambda_{\rm QCD}^{-1}$, where $r_{\bar{Q}Q}$ is the size of the heavy quarkonium. Then, at large distances, the operator for a gluon emission from a $\bar{Q}Q$ quarkonium takes the form

$$H_{\rm int} \approx -\frac{1}{2} \xi_a \vec{r} \cdot \vec{E}^a, \tag{7}$$

where $\xi^a = t_1^a - t_2^a$ is the difference between the SU(3) colour generators acting on the quark $Q$ and antiquark $\bar{Q}$, $\vec{r}$ is the relative position in the $\bar{Q}Q$ pair, and $\vec{E}^a$ is the chromoelectric field. Then the amplitude of a dipion transition between two heavy quarkonia $A$ and $B$ reads

$$\mathcal{M}(A \to B\pi\pi) = \alpha_{AB} \langle \pi\pi | \vec{E}^a \cdot \vec{E}^a | 0 \rangle, \tag{8}$$

where the effective coupling (chromopolarisability) is defined as [10]

$$\alpha_{AB} = \frac{1}{48} \langle B | \xi^a r_i G_O r_i \xi^a | A \rangle, \tag{9}$$

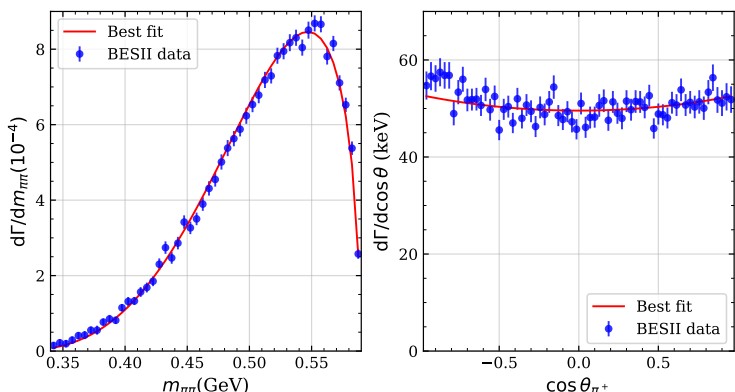

Figure 6: The fit to the BESII data on the $\psi(2S) \to J/\psi\pi^+\pi^-$ transition [11] with $\chi^2/\text{dof}=1.1$. Adapted from Ref. [3] under attribution 4.0 International license (CC BY 4.0).

with $G_O$ for the Green's function of the $\bar{Q}Q$ system in the colour-octet representation. Therefore, we resort to a two-step procedure to derive the strength of the soft-gluon exchange potential between two $J/\psi$'s. At the first stage, we extract the off-diagonal chromopolarisability $\alpha_{\psi(2S)J/\psi}$ from the BES III experimental data [11] on the dipion transition $\psi(2S) \to J/\psi\pi\pi$ (see Fig. 6) within an approach with a proper account for the final state interaction between pions and kaons that gives the value [3]

$$|\alpha_{\psi(2S)J/\psi}| \approx 1.81 \text{ GeV}^{-3}. \tag{10}$$

Then, as the second step, we use the value (10) to estimate the diagonal chromopolarisability $\alpha_{J/\psi J/\psi}$ as

$$\alpha_{J/\psi J/\psi} = \xi\alpha_{\psi(2S)J/\psi}, \tag{11}$$

where $\xi > 1$ and, according to the findings of Ref. [3], it is natural to expect $1 \lesssim \xi \lesssim 3$ or larger.

### 3.3 Potential

With the estimate of the diagonal chromopolarisability $\alpha_{J/\psi J/\psi}$ obtained in the previous chapter we are in a position to study the interaction potential between two $J/\psi$'s. To this end we employ a dispersive approach to write

$$V_{\text{tot}}(r,\Lambda) = V_\pi(r,\Lambda) + V_K(r,\Lambda) = V_{\text{CT}}(r,\Lambda) + V_{\text{exch}}(r,\Lambda), \tag{12}$$

where $V_\pi$ and $V_K$ are the two-pion and two-kaon potentials, respectively (see Fig. 7), while the contact term and long-range exchange potential read

$$V_{\text{CT}}(q,\Lambda) = \text{Const} \times F(q^2/\Lambda^2) \tag{13}$$

and

$$V_{\text{exch}}(r,\Lambda) = -\frac{1}{4\pi M_{J/\psi}^2} \int \frac{d^3q}{(2\pi)^3} e^{i\vec{q}\cdot\vec{r}} \int_{4m_\pi^2}^\infty d\mu^2 \frac{\text{Im}\mathcal{M}_{J/\psi J/\psi}(\mu^2)}{\mu^2+q^2} F\left(\frac{q^2+\mu^2}{\Lambda^2}\right), \tag{14}$$

respectively, with $\mathcal{M}_{J/\psi J/\psi} \propto \alpha_{J/\psi J/\psi}^2$ for the amplitude of the $J/\psi$-$J/\psi$ scattering through the soft gluon exchanges. Here $F(q^2/\Lambda^2)$ is a suitable form factor used to regularise the short-range behaviour of the potential. The results demonstrate only a weak dependence on the

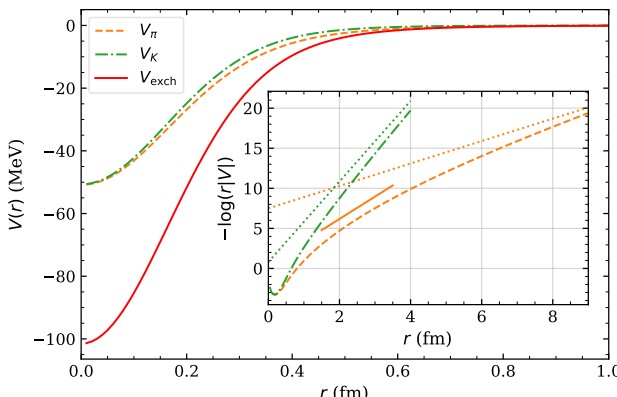

Figure 7: The behaviour of the regularised potentials $V_\pi(r, \Lambda)$ and $V_K(r, \Lambda)$ (see Eq. (12)) as functions of $r$ for the cut-off $\Lambda = 2$ GeV. Adapted from Ref. [3] under attribution 4.0 International license (CC BY 4.0).

particular form of the form factor for which we finally choose a Gaussian form (see Ref. [3] for further details). Acceptable values of the cut-off $\Lambda$ consistent with the developed approach to the interaction between $J/\psi$'s lie in the range, roughly, from 1 to 1.5 GeV [3].

It is important to notice that, in the system at hand, there are no sources for $V_{CT}$ to provide a contribution larger than that from the pion/kaon exchanges since the exchanges by soft gluons (light mesons) are OZI suppressed while exchanges by charmonia are suppressed as $\Lambda_{QCD}^2/m_c^2$. Therefore, it is natural to expect that

$$R \equiv \frac{V_{exch}^S(k' = 0, k = 0, \Lambda)}{V_{tot}^S(k' = 0, k = 0, \Lambda)} \gtrsim \frac{1}{2}. \tag{15}$$

Therefore, the answer to the question whether or not soft gluon exchanges have power to produce a near-threshold pole in the double-$J/\psi$ system amounts to a possibility to reconcile such a pole on the physical (bound state) or unphysical (virtual state) Riemann sheet with the set of constraints

$$1.0 \text{ GeV} \lesssim \Lambda \lesssim 1.5 \text{ GeV}, \quad 1 \lesssim \xi \lesssim 3, \quad R \gtrsim 0.5. \tag{16}$$

The results of our investigations are visualised in Fig. 8. To arrive at them we fix particular values of $\Lambda$ and $\xi$ consistent with Eq. (16) and, by tuning the contact potential $V_{CT}$ (effectively, the ratio $R$), ensure that the Lippmann-Schwinger equation

$$T(E; k', k) = V_{tot}^S(k', k, \Lambda) + \int \frac{d^3 l}{(2\pi)^3} \frac{V_{tot}^S(k', l, \Lambda) T(E; l, k)}{E - l^2/M_{J/\psi} + i\epsilon}, \tag{17}$$

where

$$V_{tot}^S(k', k, \Lambda) = \langle V_{tot}(\vec{k} - \vec{k}', \Lambda) \rangle_{\vec{n}'} = V_{CT}^S(k', k, \Lambda) + V_{exch}^S(k', k, \Lambda), \tag{18}$$

possesses a bound (solid line in Fig. 8) or virtual (dashed line in Fig. 8) state solution with a given binding energy $E_{pole}$ (we consider $E_{pole} = 1$ and 5 MeV). A large overlap of the corresponding bands found for $\xi = 2$ and 3 with the shaded rectangular regions in the upper left corner of the plots for both $E_{pole} = 1$ and 5 MeV implies that the existence of a molecular pole near the double-$J/\psi$ threshold is consistent with our knowledge on hadron-hadron interactions at low energies.



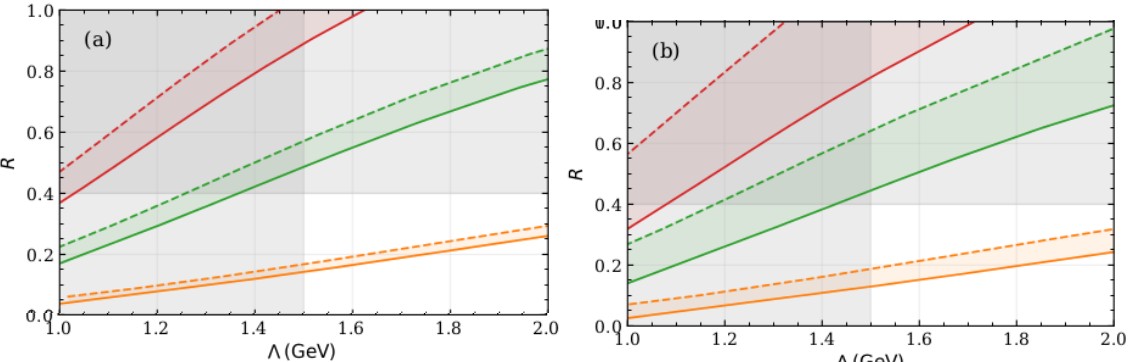

Figure 8: The dependence of the ratio $R$ from Eq. (15) on the cut-off $\Lambda$ for $E_{\text{pole}} = 1$ MeV (left plot) and $E_{\text{pole}} = 5$ MeV (right plot) below the double-$J/\psi$ threshold. For $\xi = 1$ (yellow), $\xi = 2$ (green) and $\xi = 3$ (red) the shaded band between the solid and dashed lines of the same colour corresponds to solutions consistent with a near-threshold pole on the physical or unphysical Riemann sheet residing within $E_{\text{pole}}$ from the di-$J/\psi$ threshold. Adapted from Ref. [3] under attribution 4.0 International license (CC BY 4.0).

## 4 Conclusions

The discovery of the $X(3872)$ in 2003 by the Belle Collaboration [12] started a new era in the physics of hadrons with heavy quarks. Recent data on the double-$J/\psi$ production in proton-proton collisions provided by the LHCb Collaboration [1] opened a new chapter in this book. From the theoretical analysis which respects unitarity and approximate but rather accurate HQSS we conclude that these data are consistent with a coupled-channel description, and even minimalistic models provide a good description of the data. Further experimental tests which could allow one to better constrain the theoretical models and distinguish between them include measurements in the complementary double-$\eta_c$ and $\psi(2S)J/\psi$ charmonium channels and double-$\Upsilon$ bottomonium channel. Lattice simulations of the double-$J/\psi$ and double-$\eta_c$ scattering could provide an independent test. Also, our approach predicts the $P$-wave $\pi$-$J/\psi$ scattering amplitude in the form

$$\mathcal{M}_1[J/\psi\pi \to J/\psi\pi] = 8\pi(M_{J/\psi} + m_\pi)k^2 a_1, \quad a_1 \simeq -(0.2 \sim 0.6)\,\text{GeV}^{-3}, \tag{19}$$

that could potentially be verified on the lattice, too.

From the data analysis performed we conclude that the position of the poles of the amplitude lying above the double-$J/\psi$ threshold is very vaguely fixed by the present data, however all models employed support the existence of a state with the quantum numbers $J^{PC} = 0^{++}$ or $2^{++}$ near the double-$J/\psi$ threshold. Parameters of the effective range expansion extracted from the fit to the data demonstrate that its molecular nature is plausible and compatible with the data. Thus models for the $J/\psi$-$J/\psi$ binding are welcome to investigate the nature of this proposed state $X(6200)$. In particular, we demonstrate that the existence of such a molecular pole near the double-$J/\psi$ threshold is indeed consistent with our knowledge on low-energy hadron-hadron interactions.

**Funding information**   This work was supported in part by the Chinese Academy of Sciences (CAS) under Grants No. XDPB15, No. XDB34030000, and No. QYZDB-SSW-SYS013, by the National Natural Science Foundation of China (NSFC) under Grants No. 11835015, No. 12047503 and No. 11961141012, and by the NSFC and the Deutsche Forschungsgemeinschaft (DFG) through the funds provided to the Sino-German Collaborative Research Center

"Symmetries and the Emergence of Structure in QCD" (NSFC Grant No. 12070131001, DFG Project-ID 196253076 – TRR110).

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
