# Peer review of "Double-J/ψ system in the spotlight of recent LHCb data"

_SciPost Physics Proceedings, doi:SciPost Phys. Proc. 6, 007 (2022)_

## Round 1 · Referee Report · Anonymous (Referee 1) · 2022-3-30

Report

Phenomenological importance of this study consists in finding significant ($5\sigma$) evidence for the existence of a resonant structure (X(6200)) in double $J/\psi$ system. The authors analyzed scattering data over the range $6.2<M_X<7.2$ GeV, where a very thorough model-dependent analysis is needed. The authors performed a scrupulous analysis in a minimalistic approach using a couple-channel model and found that X(6200) is more like a $J/\psi-J/\psi$ molecule rather than a tetraquark and estimated the interaction potential in $J/\psi-J/\psi$ systems as well as chromopolarizability. It should also be noticed that this study involves a phenomenological request to compute heavy-quarkonia scattering amplitude on a lattice.

---

## Editorial Decision

published